# Sustainable PHBH–Alumina Nanowire Nanocomposites: Properties and Life Cycle Assessment

**DOI:** 10.3390/polym14225033

**Published:** 2022-11-20

**Authors:** Julen Ibarretxe, Laura Alonso, Nora Aranburu, Gonzalo Guerrica-Echevarría, Amaia Orbea, Maider Iturrondobeitia

**Affiliations:** 1eMERG Research Group, School of Engineering of Bilbao, Building II-I, University of the Basque Country (UPV/EHU), Rafael Moreno Pitxitxi 3, 48013 Bilbao, Spain; 2Life Cycle Thinking Group, Department of Graphic Design and Engineering Projects, University of the Basque Country (UPV/EHU), Plaza Ingeniero Torres Quevedo 1, 48013 Bilbao, Spain; 3POLYMAT and Department of Polymer and Advanced Materials: Physics, Chemistry and Technology, Faculty of Chemistry, University of the Basque Country UPV/EHU, Paseo Manuel de Lardizabal 3, 20018 Donostia-San Sebastián, Spain; 4CBET Research Group, Department of Zoology and Animal Cell Biology, Research Centre for Experimental Marine Biology and Biotechnology PiE and Science and Technology Faculty, University of the Basque Country (UPV/EHU). Sarriena z/g, 48940 Leioa, Spain

**Keywords:** biopolymers, biodegradable, PHBH, bio composites, life cycle assessment

## Abstract

Poly(3-hydroxybutyrate-co-3-hydroxyhexanoate) (PHBH) is a bio-based polyester with the potential to replace some common polymers of fossil origin. However, PHBH presents serious limitations, such as low stiffness, tendency to undergo crystallization over long time periods and low resistance to thermal degradation during processing. In this work, we studied the use of alumina nanowires to generate PHBH–alumina nanocomposites, modifying the properties of PHBH to improve its usability. Solvent casting and melt blending were used to produce the nanocomposites. Then, their physicochemical properties and aquatic toxicity were measured. Finally, LCA was used to evaluate and compare the environmental impacts of several scenarios relevant to the processing and end of life (EoL) conditions of PHBHs. It was observed that, at low concentrations (3 wt.%), the alumina nanowires have a small positive impact on the stiffness and thermal degradation for the samples. However, for higher concentrations, the observed effects differed for each of the applied processing techniques (solvent casting or melt blending). The toxicity measurements showed that PHBH alone and in combination with alumina nanowires (10 wt.%) did not produce any impact on the survival of brine shrimp larvae after 24 and 48 h of exposure. The 18 impact categories evaluated by LCA allowed defining the most environmentally friendly conditions for the processing and EoL of PHBHs, and comparing the PHBH-related impacts to those of some of the most common fossil-based plastics. It was concluded that the preferable processing technique for PHBH is melt blending and that PHBH is unquestionably more environmentally friendly than every other analyzed plastic.

## 1. Introduction

The marine pollution produced by plastic materials is considered one of the greatest dangers of the 21st century, as it seriously threatens plant and animal species and several economic sectors, such as tourism, shipping and fishing. As a partial solution, the use of biodegradable plastics has been largely studied, supported and criticized. In this work, bioplastic-based nanocomposites, their processing and end of life have been studied thermo-physically and from an environmental point of view.

Polyhydroxy alkanoates (PHAs) are a family of polyesters that can be obtained from renewable resources, can be biodegradable and biocompatible [1], and have physicochemical properties similar to those of some of the currently most used commodity polymers [2,3,4]. Therefore, despite their relatively high price, they have gathered the interest of the industry and the research community.

The most studied and used, and hence the best known, PHAs are poly(3-hydroxybutyrate) (PHB) and the copolymer resulting from inserting hydroxyvalerate (HV) units in the hydroxybutyrate chain, poly(3-hydroxybutyrate-co-3-hydroxyvalerate (PHBV), which are used in several commercial applications already [1,4,5]. However, their high brittleness, aging (due to slow crystallization) and high thermal degradability during processing are characteristics that limit severely the range of applications of these materials [4].

A very common method to improve the properties of thermoplastics is mixing them with particles of other materials. In the case of PHAs, this method has been successfully employed by several research groups. For example, PHAs naturally exhibit good barrier properties and many studies have centered on improving further those properties, demonstrating that adding both inorganic and organic fillers to PHB and PHBV can significantly reduce their permeability (see references in [3]). There is also a significant number of publications studying the modification of the mechanical properties of PHAs by incorporating reinforcing particles into them, often of natural, biodegradable materials, seeking the formation of fully biocompatible and biodegradable composites (see, for example, reviews [2,6]). In general, relatively moderate improvements of the mechanical properties are obtained with natural fibers/particles [6] and better results are often achieved using more common inorganic reinforcing additives, like glass fiber [7]. Besides glass fiber, other inorganic reinforcing additives that have been added to PHAs include, among others, organomodified montmorillonite (MMT) clays [8], carbon nanotubes (CNT) [9], ZnO [10], sepiolite [11], silica [12] or carbon fiber [13]. In general, those additives had an accelerating effect on the crystallization kinetics, increased the stiffness, lowered the permeability, and reduced the ductility of the material.

PHBH poly(3-hydroxybutyrate-co-3-hydroxyhexanoate) is a copolymer whose properties can be tailored, to a certain degree, by adjusting the hydroxyhexanoate content [14,15]. PHBHs have a higher ductility and toughness and a slightly improved thermal stability with respect to PHB and PHBV, but they still show the aging issue and their stiffness is significantly decreased [15]. In recent years, many studies have been carried out to improve the more limiting properties of PHBHs (i.e., increasing the elastic modulus and the crystallization rate and decreasing the thermal degradation rate), as illustrated by the yearly evolution of the number of published papers dedicated to PHBH composites shown in Figure 7 in [6]. Although usually the elongation at break is reduced by adding reinforcing particles to the matrix, Xu et al. [16] showed that it is possible to simultaneously increase the elastic modulus, the tensile strength and the elongation at break of PHBH by using a combination of MMT and acetylated cellulose nanocrystals (CNC) as reinforcing additives, at least for low contents. They used a constant amount of MMT clays (3 wt.%) and varied the CNC content between 1 and 3 wt.% and observed that, up to a 2 wt.% of CNC, the mechanical properties as well as the thermal stability (measured as the onset temperature for the degradation process observed by TGA) improved moderately. Above that CNC content, the elongation at break was reduced, and the elastic modulus and the strength seemed to have reached a maximum. Hosoda et al. also reported a simultaneous improvement of the elastic modulus, the tensile strength, the elongation at break, the toughness and the thermal stability by preparing PHBH-porous cellulose composites with 16, 32 and 45 wt.% porous cellulose [17]. The reported elastic modulus, tensile strength, elongation at break and toughness for the sample containing 32 wt.% cellulose increased by an impressive 170%, 47%, 80% and 230% with respect to the neat PHBH, respectively. For the sample containing a 45 wt.% cellulose, the reported elongation at break and toughness were lower than those of the neat PHBH. To obtain such a remarkable improvement, they first produced a cellulose network and then infiltrated it with a PHBH solution, the solvent of which was evaporated to form the final product.

In the present work, PHBH–alumina nanowire composites were prepared by solvent casting and melt mixing. Solvent casting is very convenient to probe new materials and compositions, because it allows obtaining optimal dispersions even when working with difficult to disperse additives (such as nanoparticles). For instance, it is possible to sonicate the materials as required. On the other hand, melt mixing is the industry standard for thermoplastics, and hence, it is relevant to assess the feasibility of the developed composites by this technique. The aim of this work is studying the effect of adding alumina nanowires to PHBH on its mechanical properties and thermal stability. Alumina is an inert and abundant material that, to the best knowledge of the authors, has never been used to reinforce PHAs, but it has been successfully used to improve the mechanical properties of other polymers [18,19]. Concerning the size of the selected reinforcing particles, it is commonly accepted that nanoreinforcements have the potential to result in similar modification of the properties of the matrix than microreinforcements at significantly lower concentrations. Concerning the shape, Xie et al. showed that a larger shape factor of SiO_2_ particles used to modify PHBH resulted in a larger impact on the mechanical properties of the composite [20], and similar results were found by Johnsen et al. in the epoxy–alumina systems they studied [19], supporting the selection of a fibrous reinforcement.

The recent interest in PHBH (and PHAs in general) is clearly driven by environmental and sustainability reasons. Therefore, the authors consider that it is very relevant to perform an assessment of the environmental impacts of the developed composites in comparison with their commercial petrochemical based counterparts. Hence, an acute toxicity assay using brine shrimp larvae as test organisms and a Life Cycle Assessment (LCA) of the developed materials is included in this paper, as well as a sensitivity analysis to assess the effects of several alternatives (in terms of different energy sources, reuse of waste material and various reinforcing nanoparticles) to the experimentally employed processes.

## 2. Materials and Methods

Kaneka PHBH grade X131A (which has a 6 mol.% hydroxyhexanoate and a molar mass of 6.1 × 10^5^ g/mol) was reinforced with Sigma Aldrich alumina nanowires (product number 551,643, with a diameter of 2–6 nm and a length of 200–400 nm). To dissolve the polymer, chloroform (Sigma Aldrich ≥ 99%, containing 0.5–1.0% ethanol as stabilizer) was selected.

The samples were prepared by means of two techniques (see Figure 1): solvent casting and melt blending. For solvent casting, the polymer was dissolved in chloroform and the alumina nanowires were dispersed by stirring the mix for 40 min at 80 °C. Then, the samples were poured on glass and let dry overnight at room temperature to obtain 30–50 µm thick films. For the melt blending process, a DSM (DSM BV, The Netherlands) mini-extruder was used (at 150 rpm and 3 min of residence time at 150 °C), and then dog-bone tensile test specimens were injection molded in a DSM mini-injection molding machine.

Samples containing 0, 3, 5 and 10 wt.% of alumina nanowires were prepared using both processing techniques.

The aging (evolution of the crystallinity) of the melt blended PHBH and its composites was analyzed by Differential Scanning Calorimetry (DSC) on a Mettler-Toledo DSC1, for which samples of approximately 5 mg were used. First, all samples were kept at 180 °C for 5 min to erase their thermal history, and then they were cooled down to room temperature (20 °C) at 10 °C/min. Then, each sample was analyzed at the corresponding time (immediately after the cool down step, or after 1 day, 2 days, 4 days, 7 days, 14 days or 30 days) in heating (from 20 °C to 180 °C at 10 °C/min), from which the final melting peak temperature and melting enthalpy were measured. All samples were kept at room temperature between measurements. The melting enthalpy was measured using the extrapolation from the melt method [21].

Thermo Gravimetric Analysis (TGA) was carried out to assess the thermal stability of the composites as well as the processed neat polymer on a Mettler-Toledo TGA/DSC1. For the measurements, the samples were heated from 25 °C to 500 °C at 10 °C/min under nitrogen atmosphere and their mass was recorded. The degradation onset (temperature at which 10% of the mass of the polymer is lost) and offset (temperature at which 90% of the mass of the polymer is lost) temperatures were determined as parameters to evaluate the resistance to degradation. An amount of 5–10 mg of each sample was analyzed by TGA.

The mechanical characterization of the melt blended samples was performed by DMA in tensile mode, applying a 0.05% strain at a frequency of 1 Hz and a heating rate of 4 °C/min. The data were acquired in temperature ramps from −30 °C to 75 °C. A single sample obtained from the central section of the injection molded dog-bone specimens was tested per material. The dimensions of the parts used for DMA were 30 mm × 2 mm × 5 mm, approximately. The tests were carried out in a TA Q800 viscoelastometer.The mechanical characterization of the solvent cast films was performed by conventional tensile tests on an Instron 4206 universal tensile test machine. For each material, 8 to 10 square samples (having a width of 5–6 mm and a length of 30 mm, approximately) were cut from the films and tested at a speed of 2 mm/min.

To evaluate the size dispersion of the nanoparticles, Transmission Electron Microscopy (TEM) was used. Thin slices (approximately 100 nm thick) were cut from the injection molded dog-bone specimens on a Leica ultra-cryomicrotome. The TEM images were acquired at a pixel size of 1.33 nm/pixel and the micrographs were analyzed (filtered, binarized and quantified) using Fiji [22]. Several images (4 or 5 4.7 µm × 4.7 µm images per sample) were acquired and analyzed. The particle area was quantified and only particles having areas above 25 nm^2^ were considered to avoid incorporating in the data artifacts produced by the image processing. To estimate the error of the computed mean, the standard error was used.
(1)$$SE=s/\sqrt{n}$$
where *s* is the standard deviation of the sample and *n* the number of particles. The solvent cast materials were prepared for TEM as well. They were embedded in an epoxy resin that cured at room temperature and then, applying the same procedure as for the melt blended materials, thin slices were obtained. The resulting slices presented several issues (artifacts) and it was opted not to use them for the quantification.

Additionally, the environmental viability of PHBH nanocomposites was assessed by LCA according to ISO14040, as follows:-Goal and Scope:

The quantification of the environmental impacts was calculated in order to evaluate the real feasibility of using PHBH polymer in the industry to replace petrochemical-based and non-degradable conventional polymers. The functional unit selected for the LCA was *the use of 1 Kg of polymer from production to processing, use and End of Life (EoL).* Cradle to Grave boundaries were used.

The studied cases were the following:PHBH polymer: production, processing by melt blending and EoL composting.PHBH polymer: production, processing by melt blending and EoL biodegradation in seawater.Petroleum-based polyamide (PA), polypropylene (PP), polyethylene terephthalate (PET) and polyethylene (PE) production, processing by melt blending and processing and Landfill EoL.PHBH polymer: production, processing by melt blending (electricity mix), EoL biodegradation in seawater.PHBH polymer: production, processing by solvent casting (electricity mix), EoL biodegradation in seawater.PHBH polymer: production, processing by melt blending (renewable energy from wind), EoL biodegradation in seawater.PHBH polymer: production, processing by solvent casting (recirculation of chloroform), EoL biodegradation in seawater.PHBH polymer: production, processing by solvent casting (recirculation of chloroform + renewable energy from wind), EoL biodegradation in seawater.PHBH nanocomposites containing 3 wt.% of alumina, clay, graphene and cellulose (chemical and natural), obtained by solvent casting (electricity mix).


The flow diagrams of the studied scenarios are shown in Figure 2.

-Inventory:

For this section, data directly measured in the laboratory were used for the melt blending and solvent casting processes. The production of PHBH was modelized from bibliography data [23]. As for the EoL options: the landfill process from ecoinvent 3.8 was used, and composting and biodegradation in seawater were modelized using data from [24] and [25], respectively. The data introduced in the LCA software were obtained from the ecoinvent 3.8 and GaBi Bioplastics 2019 databases.

All the inventories are summarized in Appendix A.

-Environmental impact assessment:

For the quantification of the environmental impacts, the OpenLCA 1.11 software and the Recipe 2016 Midpoint (E) methodology were used. This methodology provides an in-depth analysis including 18 different categories of environmental impacts.

Finally, toxicity assays in brine shrimp larvae and chemical analyses of aluminum concentration were carried out. For the toxicity assays, pieces of 1 cm^2^ of the cast films of neat PHBH and PHBH containing the highest content of alumina nanowires (10 wt.%) were prepared.

Brine shrimp larvae were obtained by adding dry cysts (Artemia Koral GmbH, Germany) to 30‰ salt water prepared from commercial salt (Sera, Heinsberg, Germany), using a hatching dish in a temperature-controlled room at 26 °C and under continuous illumination. Larvae of 24 h post hatching (hph) were harvested and selected for the test under a stereoscopic microscope (Nikon smz800, Tokyo, Japan). Larvae immobilization at 24 and 48 h of exposure was used as a criterion for acute toxicity [26]. At least 5 individuals were transferred and maintained for 48 h in each well of the 24-well polystyrene microplates containing 2 mL of exposure medium. Six wells were used per treatment. The following treatments were considered: (1) 30‰ salt water as a negative control; (2) neat PHBH; (3) PHBH–alumina (10 wt.%); and (4) alumina at 0.3354 mg/mL as the equivalent alumina content present in the PHBH–alumina films. One day prior to the test, the microplate was filled with the corresponding treatments. After 24 and 48 h of exposure, the number of immobilized larvae was recorded. A larva was considered as immobile when it was not able to move after shaking the plate.

In order to monitor the aluminum concentration in the treatments, before adding the larvae, 50 µL of medium of each well were sampled to generate 3 samples of 100 µL for each treatment. In the case of the wells containing only alumina, precipitated particles were resuspended by pipetting before sample collection. Exposure medium sample collection was repeated after 24 h of exposure but, in this case, alumina nanoparticles were not resuspended to avoid causing any harm to the larvae in those samples. An amount of 300 µL of concentrated nitric acid was added to each sample. Before the aluminum concentration was measured by ICP-OES, the samples were diluted 1:100 with ultrapure water to avoid interference of the salt water with the ICP-OES (Agilent 5100), which used a quartz Meinhard concentric nebulizer and a Scott-type spray chamber. Working standard solutions of Al were prepared immediately prior to their use, by stepwise dilution of certified standard multi-element solution (1000 mg L^−1^) (Merck, Darmstadt, Germany), with HNO_3_ 1.0% *v*/*v*.

## 3. Results

### 3.1. Evolution of Crystallinity over Time

Figure 3 presents the evolution over time of the melting enthalpy per gram of polymer and melting peak temperature of the materials prepared by melt blending, as obtained by DSC. Although there is a significant degree of fluctuation in the enthalpy data due to shape of the melting curves, which were not easy to analyze univocally despite using the extrapolation from the melt method for that [21], the results show that the melting enthalpy per gram of polymer is stable over the studied period of time, even for the neat PHBH. Actually, there is a very small correlation between the average enthalpy for each day and the time, but the mentioned fluctuation in the data could be the cause for it. Similarly, the melting peak temperature remains very stable over time, which indicates that no annealing is taking place at room temperature.

Moreover, the data seem to show that the alumina nanowires do not cause additional crystallinity in the composites over time, as evidenced by the similar melting enthalpies measured for all samples after 30 days, indicating that the alumina has no nucleating effect. Additionally, the melting peak temperatures are similar for all the studied materials, which could indicate that there is no significant difference between them regarding the crystal structure.

The crystallinity after 30 days is around 31% for all samples (calculated using 146 J/g a reference melting enthalpy for a 100% crystalline PHB [27]).

### 3.2. Thermal Stability

TGA was used to assess the resistance to thermal degradation of the prepared composites and the results are summarized in Figure 4. The full TGA curves have been included in the Appendix A.

The TGA data show that the effect of small amounts of alumina particles is in general positive, as evidenced by the overall increased degradation onset and offset temperatures when up to a 3 wt.% is added to the polymer by either of the used processing methods. In melt blended, the improvement is not very significant. Moreover, in melt blended samples, increasing further the amount of alumina results in lower degradation temperatures. On the other hand, in the case of the solvent cast samples, increasing the amount of alumina produces a delay in both onset and offset temperatures up to the largest particle content used here.

The chosen processing method seems to have a significantly higher impact on the thermal stability of the produced materials than the composition itself. Indeed, the solvent cast materials display a greater tendency to undergo thermal degradation than the melt blended ones.

### 3.3. Mechanical Properties

The DMA-measured storage moduli and tan δ values as a function of temperature of the samples prepared by melt blending are shown in Figure 5. The storage moduli for the room temperature range can be seen in more detail in the inset. The nanowires have a reinforcing effect on the matrix, as evidenced by the increase in the storage modulus as a function of the alumina content. However, the temperature seems to play a significant role too. At room temperature and above, there is very little difference between the samples containing 3 wt.% and 5 wt.% alumina, whereas at lower temperatures, their moduli differ much more. The glass transition temperature (peak temperature of the tan δ curve) is very similar for all the compositions studied, with a value of approximately 25.5 °C.

The elastic modulus, the elongation at break and the tensile strength of the solvent cast samples, as measured by tensile tests, are presented in Table 1. DMA was not selected for these measurements due to the thin geometry of the films. The corresponding tensile test curves are included in the Appendix A.

For the sake of clarity, the results given in Table 1 have also been plotted, in Figure 6, relative to the values of the sample without alumina.

The tensile test results for the solvent cast samples show that the initial addition of alumina nanowires (up to 5 wt.%) has a positive effect on both the elastic modulus and the tensile strength. However, increasing further the amount of alumina in the samples (to a 10 wt.%) results in a decrease in those properties. Concerning the elongation at break, there is a significant decrease already for the smallest alumina content, but no further reduction of the elongation is observed at higher contents. These trends (increased modulus and strength and decreased elongation at break) are commonly observed for nanocomposites, as mentioned in the introduction.

### 3.4. Microstructural Analysis

TEM was used to analyze qualitatively and quantitatively the microstructures of the alumina nanoparticles and the prepared composites. The TEM micrographs of the reinforcing particles show the expected nanowire shape and aggregation state (see Figure 7). As for the size, the average diameter and length were obtained from the measurements of over 50 particles, resulting in a 5 ± 1 nm diameter and a 17 ± 4 nm length. These lengths are very far from the nominal 200–400 nm announced by the provider of the nanowires.

The melt blended PHBH–alumina nanocomposites were analyzed by TEM as well. The melt blended materials were selected over the solvent cast ones because their preparation for TEM was simpler. An example of each of the composites is shown in Figure 8.

All observed particles are in the submicrometer range (the largest ones, which are agglomerates of smaller particles, are below 0.5 μm in diameter), and a significant fraction of them are smaller, nanometric particles. Both large and small particles are observed in all three nanocomposites.

In order to determine the degree of dispersion as a function of the amount of incorporated reinforcing particles, the micrographs of the composites were quantitatively analyzed. The results are shown in Figure 9 and Table 2.

Automated Image Analysis [28] was used to preprocess and binarize the images, and to compute the selected microstructural descriptors—particle density and average particle size (area)—as shown in Table 2. A total of 3405, 7934 and 17,103 particles were measured for the nanocomposites containing 3 wt.%, 5 wt.% and 10 wt.% alumina, respectively, which resulted in 3 nm^2^ or lower standard errors. The standard deviation of the measured sizes is significant, as could be anticipated by the features observed in the TEM images. Moreover, the orientation of the nanowires was not considered here, despite its very relevant effect on the obtained figures. A nanowire oriented perpendicular to the plane of the image would have a projected size of around 20 nm^2^ (since the diameter of the nanowires is roughly 5 nm, and assuming a cylindrical shape), whereas an equal length nanowire oriented parallel to the plane of the image would have a much larger projected size. Therefore, the presented figures will be taken as a semi-quantitative estimation of the actual size of the particles, and will only be used for comparison purposes.

As expected, the data show that the density of the measured particles increases as the alumina content increases. On the other hand, the average particle size decreases when more alumina is added, which contradicts previous reports [28] and seems counterintuitive when checking the TEM images. Actually, since the number-average particle size is reported here, and the influence of the smaller particles on that parameter is very significant, even if larger particles and in larger quantities are present as the amount of alumina is increased, the average size will decrease if the number of small particles increases to a larger degree too, which is the case here. In any case, the images clearly show that the number of large aggregates and their size increase with the alumina content.

The higher magnification TEM micrograph presented in Figure 10 shows aggregates with a morphology of not-dispersed nanowires and a size of approximately 100 nm.

Finally, the TEM micrographs of the composites prepared by solvent casting and melt blending were compared (see Figure 11, where only the 3 wt.% samples are displayed). The images show a similar morphology for both materials, and the size range of the particles seems equal too. However, the number of small particles visible in the melt blended material seems significantly higher as compared to that of the material obtained by solvent casting.

### 3.5. Environmental Impact Assessment by Life Cycle Assessment

The results obtained from the LCA, using the Recipe 2016 Midpoint E methodology, are summarized in Figure 12 and Appendix A. For better comprehension, a more detailed analysis is carried out in the following subsections.

#### 3.5.1. PHBH vs. Petrochemical Based Conventional Polymers

In this section, environmental impacts of the scenarios “a”, “b” and “c” are analyzed. For scenarios “a” and “b”, PHBH is the selected polymer, and composting and biodegradation in seawater are the EoL conditions, respectively, whereas for scenario “c”, petrochemical-origin polymers (PA, PP, PET, PP, and PET) and landfill disposal as EoL have been used.

In general, as Figure 12 shows, 14 out of the 18 categories indicate that the most polluting scenario is that related to the production and end of life of PA (of petrochemical origin). For the category of “Land Use”, the composting scenario has the greatest influence. To clarify that result, it should be mentioned that the land use is not taken into account in the landfill end of life scenario in ecoinvent 3.8; in fact, the process is carried out in industrial units. For “stratospheric ozone depletion” and “fresh water ecotoxicity”, the environmental impacts are larger for PHBH production and end of life. This is due to the emissions (methane, for instance) during its biodegradation process, which happens in open land and therefore the emissions are emitted to the air and ground directly.

Due to the current climate crisis and the specific product that is under study here, a few of those 18 environmental impact categories may be considered more relevant than the others. A more detailed analysis of those categories will be provided next.

The global warming category is a relevant impact indicator with regard to the transition to a low-carbon economy and to the fulfillment of the Paris Agreement, according to which a 55% emission reduction must be achieved by 2030. Global warming is an indicator of the amount of CO_2_ emitted, which is the cause of the increase in the Earth’s temperature. Figure 13 shows the global warming category results for cases “a”, “b” and “c”.

The use of petrochemical-based and non-biodegradable PA has severe effects in this category, in which the resulting impacts increase from 1.66 Kg CO_2_-eq. for PHBH, up to 24.5 Kg CO_2_-eq. for PA (which scores the highest global warming impact among petrol-based polymers). Up to 50% of the contribution to the global warming effect comes from the production process of PA, where the emissions of carbon dioxide and dinitrogen monoxide are high.

Most of the LCA studies found in the literature and related to PHB are linked to the production step. A summary of some of the results found in the literature is shown in Table 3.

The results obtained in the present work (see the last row in Table 3) are in the same range and come into agreement with those found in the literature for the global warming category. On the contrary, AP and EP category results from this study are below the rest. This fact can be attributed to different methodologies used for the calculation of the environmental categories, as well as the end of life (biodegradation), which is included in this research work. However, with respect to those previously reported by other groups, this study broadens the system boundaries used for the LCA, including two different processing techniques and different end of life options. Moreover, we also provide 15 impact categories in addition to those shown in Table 2, which give a wider and deeper insight into the studied systems.

Next, four impact categories related to toxicity during the use and EoL of the studied materials (namely, the fine particulate matter formation category, the marine ecotoxicity, the human carcinogenic toxicity and the terrestrial ecotoxicity) will be discussed together. Terrestrial and marine ecotoxicities are considered relevant aspects given the possible EoL of the materials studied here, which often end up in oceans or landfills or under composting conditions. Fine particulate matter production and human carcinogenic toxicity were selected for this in-detail analysis due to the direct relationship between new materials and technologies and human health. The results for those four categories are shown in Figure 14.

Marine ecotoxicity provides a measurement of the effect on the health of the oceans and, according to the LCA results, it can be reduced from 3500 kg 1,4-DCB for PA to 400 kg 1,4-DCB for PHBH. Terrestrial ecotoxicity is an indicator of the negative impact on the well-being of the terrestrial ecosystems. This category measures the negative effects on the flora and fauna in our lands, which directly affects the entire trophic chain. From the corresponding data of Figure 14, it can be concluded that it is significantly beneficial to use PHBH instead of PA in terms of terrestrial ecotoxicity, which is reduced from 11 kg 1,4-DCB (PA) to 4 kg 1,4-DCB (PHBH). Fine particulate matter production category indicates the number of particles emitted (which can generate health problems in humans), and it would be reduced by roughly a factor of 10 when using PHBH instead of PA. Human carcinogenic toxicity measures the potential for causing cancer in humans. In this case, the use of PHBH induces a reduction in this impact from 14 kg 1,4-DCB for PA to 7 kg 1,4-DCB for PHBH.

In general, the main contributors to the environmental impacts of using PA are related to the production of the polymer itself. Large emissions of sulfur dioxide and nitrogen dioxide to the air, and vanadium, copper, nickel, zinc, mercury and selenium to the water and air are responsible for PA’s high impacts.

For the case of PHBH, the main contributions come from ammonia and its processing used in the production of the biopolymer.

#### 3.5.2. Sensitivity Analysis: PHBH Melt Blending vs. PHBH Solvent Casting (Scenarios “d” to “h”)

Optimizing the material processing techniques is another key issue in order to evolve to more sustainable activity. Thus, solvent casting and melt blending (extrusion) processing techniques have been used and compared in this study. In addition, a sensitivity analysis of several scenarios considering different processing technologies and electric energy sources has been carried out. The considered scenarios are presented in Appendix A, and the results can be found in Appendix A. Common strategies to reduce the environmental impacts of processing include reducing the amount of used materials and reducing processing time and temperature. In this study, however, the focus will be on the effect of the reuse of waste solvents and the selection of the energy source (electric mix or electricity from wind technology).

As expected, the solvent casting technique is not an environmentally friendly process, as the results shown in Figure 15 reveal. In this sense, solvent casting might be more suitable at laboratory scale during the early design processes of new materials. The main contribution to the environmental impacts of solvent casting stems from the use of solvents. Therefore, in order to analyze the potential benefit of reusing the solvents, a recuperation and recirculation of the main solvent, i.e., the chloroform, was simulated. The electricity consumption is the second main contributor to the environmental impacts. Hence, the use of renewable energy sources (wind energy, specifically) was evaluated as well. As can be seen in Figure 15, by including waste treatment and recirculation as well as using renewable source electricity, the environmental impacts for solvent casting are reduced by more than 60%. In any case, even after such an improvement, the melt blending option presents lower impacts.

#### 3.5.3. Sensitivity Analysis: Nanoreinforcements to Enhance the Properties (Scenario Defined in “i”)

The wide adoption of PHBHs in the industry is not immediate, since their physicochemical properties are not competitive with those of the majority of conventional polymers of fossil origin. That is the main driver for this work, which aims to improve the properties of PHBH by the addition of nanoparticles [37], as stated before. That is why alumina nanowires have been used here, taking advantage of their inert nature and their potential positive effect on the properties that need to be improved. However, other reinforcing materials could be (more) suitable to attain the sought goals. In order to compare the effect of the selected reinforcement with other, commonly used, reinforcing nanoparticles, an additional sensitivity analysis was carried out: cellulose (chemical and natural), clay nanoparticles and graphene were used in addition to the alumina nanowires for the analysis. Cellulose is a natural, organic and renewable material that can be found as nanoparticles [38]; nanoclays are natural, inorganic, laminar reinforcements [39]; and graphene is a singular carbon structure composed of a single monoatomic layer, which is conductive and has the potential to reinforce polymers [40]. The benefit of using nanoparticles lies in the small amount required to obtain improved properties. For this sensitivity analysis, an addition of 3 wt.% was selected.

As presented in Figure 16b (see Appendix A) the results clearly show that the environmental impacts are not sensitive to the addition of such small amounts of different reinforcements.

However, the LCA methodology does not take into account the size and geometry of the reinforcing particles. It is well known that nanometric particles can be harmful to the environment, as well as to humans [41].

### 3.6. Toxicity

Exposure of brine shrimp larvae of 24 hph for 24 and 48 h to films of PHBH and PHBH–alumina (10 wt.%) composites prepared by solvent casting has no impact on the survival of the organisms. After 48 h of exposure, 100% survival was registered in control animals (salt water treatment), as well as in animals exposed to both film types. In order to test whether a suspension containing the equivalent alumina amount present in the PHBH–alumina film caused any toxic impact on the organisms, larvae were also exposed to alumina (0.3354 mg/mL). In this treatment, only one individual (out of 41) died after 48 h of exposure, resulting in a survival rate of 97.5%. No mortality was detected in this treatment after 24 h of exposure.

Chemical analysis of aluminum concentration in exposure media reveals a background concentration of 35.77 ± 0.12 and 36.37 ± 0.21 ppm (at 24 and 48 h of film incubation, respectively) in the PHBH treatment, and of 36.27 ± 0.38 and 36.53 ± 0.12 ppm in the PHBH–alumina treatment, indicating that the release of Al to the exposure media is almost negligible. Al concentration measured in the media containing alumina alone before adding the test organisms (at 24 h of film incubation) reached 68.27 ± 1.91 ppm, but dropped to 37.57 ± 0.06 ppm at 48 h. The difference between these two time points is due to the sedimentation of the alumina particles at 48 h, when the sedimented particles were not resuspended to avoid interference with the test organisms. Moreover, the concentration measured at 24 h is below the nominal concentration, suggesting that the acid treatment of the samples was not strong enough to dissolve the alumina nanoparticles and make Al detectable by ICP-OES.

## 4. Discussion and Conclusions

As stated before, from the environmental point of view, PHBH is an interesting potential alternative for some of the most used commodity polymers. However, PHBHs’ tendency to undergo aging, their low stiffness and low thermal stability [4] pose a significant barrier for their adoption by the plastic industry. In this work, we studied the effect of mixing PHBH with a reinforcing material (i.e., alumina nanowires), since it is possible and expected that adding a nano-reinforcement to a polymer matrix will cause the mentioned three problematic properties to improve.

Two processing techniques were employed for the preparation of the nanocomposites, i.e., melt blending (extrusion and injection molding) and solvent casting, and the obtained samples were thermally and mechanically characterized. Three different alumina contents were studied (3, 5 and 10 wt.%) and compared to the processed, neat polymer.

In addition, an assessment of the environmental effects posed by the production and use of PHBH–alumina nanocomposites was carried out using two techniques. First, an LCA of the environmental impacts related to the production, use and EoL of PHBH was performed and compared to those of the most common commodity petrochemical-based thermoplastics. Moreover, several scenarios (by varying the processing technique and conditions, the EoL conditions and the type of energy used for the processing) were simulated in order to obtain a clear and thorough characterization of the actual environmental impacts of the different plausible scenarios. Cradle to Grave boundaries were applied for the LCAs, whereas most studies found in the literature are limited to the production of the polymer. Eighteen environmental impact categories were evaluated in the LCA in order to ensure an in-depth analysis. In addition to the LCAs, toxicity assays in brine shrimp larvae and a determination of the amount of aluminum released by the samples when submerged in water were performed.

Concerning the evolution of the crystal morphology of the samples (i.e., aging), the DSC data indicate that both the crystallinity and the crystal structure remain stable over the observation time even for the processed, neat PHBH, which suggests that this specific PHBH grade is not affected by the aging issue. In any case, very little difference, if any, exists between the composites and the neat polymer, i.e., the crystallinity and melting temperatures of all samples are very similar, which means that, unexpectedly [8,9,11], the crystallization behavior and crystal structure are not affected by the alumina particles, at least up to a 10 wt.% content. This behavior would be expected if the PHBH used contained a nucleating agent shadowing the effect of the alumina particles on the crystallization behavior. Since the PHBH used here is a commercial grade, that may well be the case.

Regarding the thermal stability (as measured by TGA), it was observed that, in general, the addition of alumina nanoparticles delayed the degradation process to higher temperatures, but the improvement was not significant (just over 10 °C increase in the best case). Here, the processing technique used made a difference. Whereas the melt blended samples displayed a maximum degradation temperature for the composite containing 3 wt.% alumina, in the case of the solvent cast samples, the degradation temperature increased monotonically with the amount of alumina (although at lower increase rates for higher alumina contents and apparently nearly reaching a maximum for the 10 wt.% alumina content). A known mechanism by which the dispersed particles delay thermal degradation is by the increase of the tortuosity for the products of the degradation [42]. Thus, by increasing the number of dispersed particles, in general, such effect should be greater too, which could partially explain the observed behavior. Concerning the smaller degradation temperatures of the melt blended samples containing 5 wt.% and 10 wt.% alumina, it is clear the alumina itself has a detrimental effect that outweighs that of the increased tortuosity (which is also limited by the poorer dispersion of the alumina for the higher contents, as measured from the TEM micrographs). Further investigation would be required to completely elucidate the actual reasons behind the observed behavior, but that lies outside the scope of this work.

The mechanical properties of the prepared samples were measured by DMA (for the melt blended materials) and tensile tests (for the solvent cast ones). Concerning the stiffness of the composites, significant increases are observed: approximately, a 60% increase in the storage modulus by adding 10 wt.% alumina by melt blending, and a 43% increase of Young’s modulus by adding 5 wt.% alumina by solvent casting. The tensile strength of the solvent cast composites is also maximum for that same alumina content with a 44% increase with respect to the strength of the neat PHBH. On the other hand, the elongation at break decreases when alumina is added to the polymer, although that effect is very strong initially, but less prominent for higher alumina contents. It is widely accepted that, in order to obtain an efficient reinforcing effect, a good dispersion of the particles is required. Although the reported particle size data as computed from the TEM images indicate that the average particle size decreases with the alumina content, it is also observed that the amount and size of aggregates increases too. These aggregates become weak spots within the material [43], causing a decrease in the stiffness and the strength. A further improvement of the mechanical properties (presumably resulting in a lower decrease of the elongation at break as well) may be possible by using compatibilizers to reduce the number and size of aggregates [18], and possibly improve the stress transfer across the interphase between the polymeric matrix and the nanoparticles. On the other hand, the addition of a new component to the composites has other ramifications (in terms of thermal stability, toxicity, biodegradability and environmental impacts) that should also be studied, but it is certainly an interesting prospect for future research.

According to the LCA, using PHBH instead of any of the petrochemical-origin plastics that have been assessed here (i.e., PA, PE, PP, and PET) results in a reduction of most of the 18 environmental impact categories that have been assessed, even if the PHBH ends up as sea waste. Concerning the PHBH itself, the impacts associated with the solvent casting technique are significantly higher than those obtained for melt blending. This negative result can be mitigated when adding solvent recovery to the solvent casting process and when energy from renewable sources is used, but solvent casting still scores remarkably worse than melt blending. In the case of the melt blended materials, which use an energy intensive technique, using energy from renewable sources has a significantly positive effect on several of the assessed impacts.

LCA was also applied to study the environmental implications of the selected reinforcing nanoparticles. Alumina was used as reinforcement in the experimental part of this work, but other materials were also considered. Those other nanoparticles may reduce the environmental impacts of the generated materials while potentially improving PHBH’s problematic material properties discussed before. The corresponding LCA results show almost identical impact values for all considered materials (PHBH and nanocomposites thereof obtained by adding 3 wt.% of alumina, cellulose, graphene and clays to the polymer) processed by solvent casting. The number of nanoparticles required to effectively improve the physicochemical properties of the matrix is small, and therefore, their weight on the total impacts is not significant. From the point of view of the environmental impacts, it can be concluded that the nanocomposites should be melt blended, preferably using green energy, and that the nature of the used reinforcement (among the studied ones) is irrelevant.

However, the performed LCA is based on the ecoinvent 3.8 database that does not take into account the effect of the nanometric dimensions of the alumina nanowires. Some studies have been already done to test the potential impact of PHB on aquatic organisms, but the experimental designs are diverse and controversial results have been reported. In this work, we studied the effect of the 48 h exposure to PHBH films produced by solvent casting on the viability of brine shrimp larvae. The results show no effect under the assayed conditions. In agreement with our results, Tanadchangsaeng and Pattanasupong (2022) investigated residual toxicity after biodegradation of PHB and observed 100% viability in brine shrimps (*Artemia franciscana*). Straub et al. studied the effect of the ingestion of PHB microparticles in the freshwater amphipod *Gammarus fossarum*. They didn’t report acute effects, but a significantly lower wet weight gain relative to the control treatments was observed. Nevertheless, the effect was lower than in the case of ingestion of a petroleum-based polymethylmethacrylate [44]. On the contrary, González-Pleiter et al. [45] reported considerable toxic effects of PHB secondary nanoplastics on three representative aquatic organisms. The growth rate of the cyanobacteria *Anabaena sp* and of the microalgae *Chlorella reinhardtii* decreased by 90 and 95%, respectively, while a significant immobilization (85%) was recorded in the crustacean *Daphnia magna*. Moreover, a significant increase in intracellular reactive oxygen species levels leading to cellular membrane damage was observed in all organisms after exposure to the PHB nanoplastics. Beiras et al. [46] compared the toxicity of PHB, LDPE, PVC and PA containing conventional and alternative additives towards microalgae (*Tisochrysis lutea*), copepods (*Acartia clausi*) and sea urchin (*Paracentrotus lividus*) embryos and claimed that the PHB toxicity is associated to the higher abundance of particles within the nanometric range found in PHB and absent in other materials. Thus, the toxicity of PHB reported by González-Pleiter et al. (2019) and Beiras et al. (2021) seems to be more related to the geometry and polymer particle size (nano) than to the polymer composition.The incorporation of alumina nanoparticles in the polymer formulation can potentially alter its environmental impact. To the best of our knowledge, there is no previous information on the aquatic toxicity of this nanocomposite. In the present work, the addition of up to 10% of alumina to the PHBH did not increase the toxicity towards brine shrimp larvae at 24 or 48 h. Moreover, exposure to alumina nanoparticles alone only caused a slight decrease of larvae viability (from 100% to 97% at 48 h). In agreement, Ates et al. [47] reported a non-significant increase of brine shrimp larvae mortality after 24 h of exposure to α- and γ-alumina nanoparticles of different sizes up to 100 mg/L. Longer exposures (96 h) increased mortality rate at the highest concentrations and caused oxidative stress as reflected in lipid peroxidation levels. In summary, direct exposure to PHBH and PHBH–alumina films produced by solvent casting in this work does not provoke acute effects on the selected model. Further studies considering particles or substances arising from biodegradation processes, aquatic species with different sensitivity and sublethal effects would be necessary for a better understanding of the environmental hazard posed by these novel formulations.

To conclude, evolving towards a more sustainable production of goods involves using innovative technologies, energy sources and materials, and trying to reach a compromise between optimum properties, costs and environmental effects. The plastic industry, which currently uses enormous amounts of petrochemical-origin plastics that often end up as waste in the environment, is a sector where innovative materials are needed. PHAs, and specifically PHBHs, are bio-based and biodegradable polymers and appear as an appealing alternative to some of the most common commercial plastics. The work presented here is a small step forward towards achieving the substitution of those petrochemical-origin plastics with bio-based plastics.

## Figures and Tables

**Figure 1 polymers-14-05033-f001:**
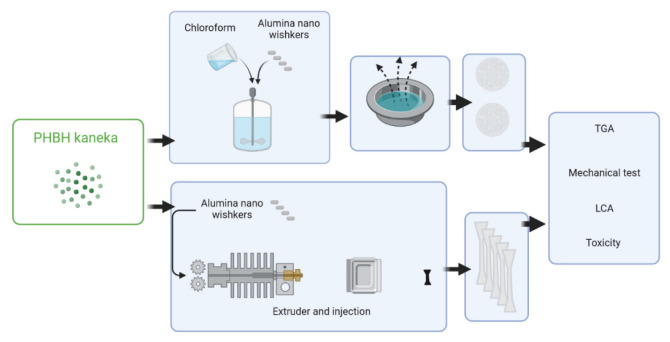
Schematic illustration of the sample production.

**Figure 2 polymers-14-05033-f002:**
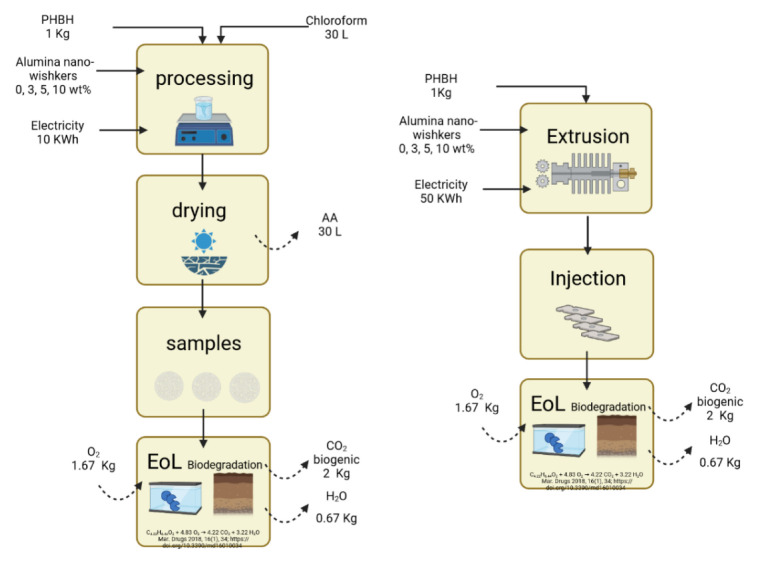
Flow diagrams used to carry out the inventory for PHBH nanocomposites.

**Figure 3 polymers-14-05033-f003:**
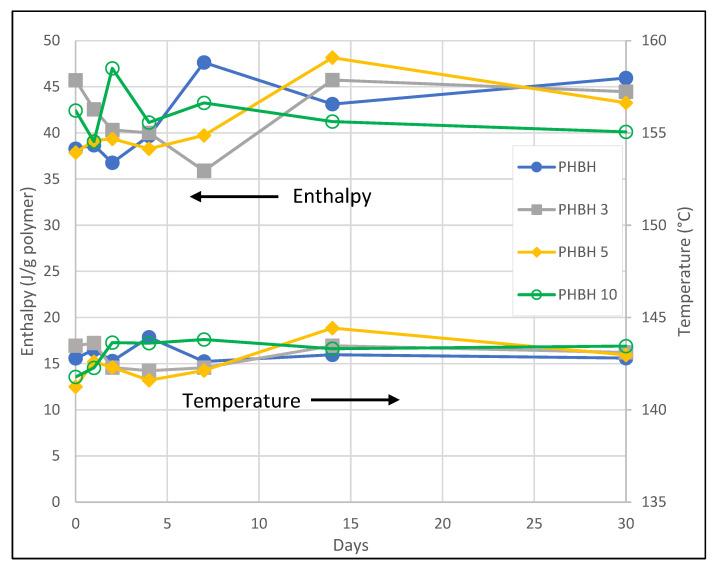
Evolution of the enthalpy of melting and the melting peak temperature over time for the neat PHBH and the composites processed by melt blending. The enthalpies are given per gram of polymer.

**Figure 4 polymers-14-05033-f004:**
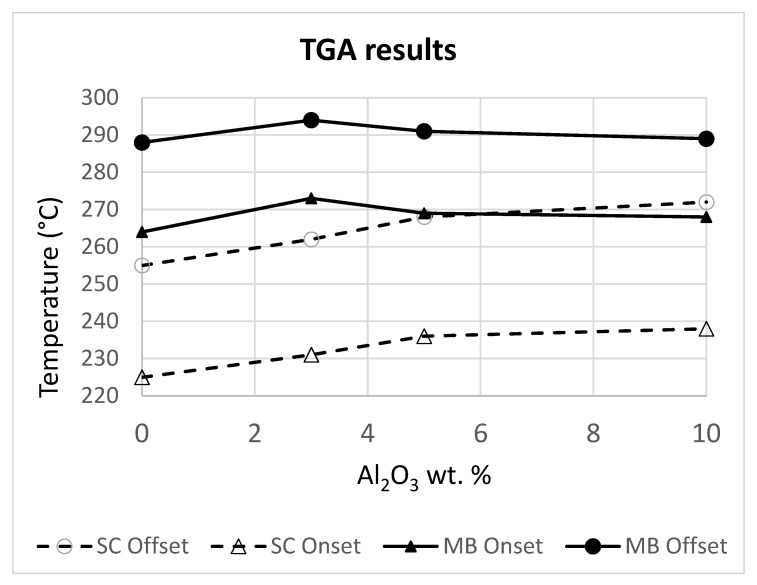
Results of the TGA measurements for the melt blended (MB) and solvent cast (SC) neat PHBH and its composites. The degradation onset and offset temperatures are shown.

**Figure 5 polymers-14-05033-f005:**
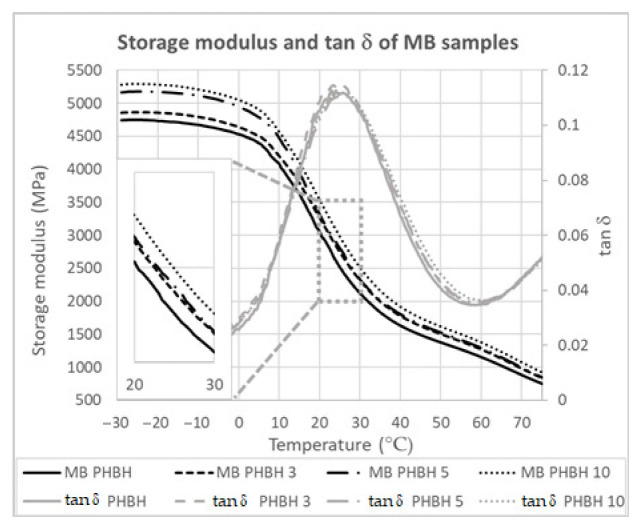
DMA- measured storage modulus and tan δ curves of the samples prepared by melt blending. An inset has been added to show in more detail the storage modulus values for the temperature range between 20 °C and 30 °C.

**Figure 6 polymers-14-05033-f006:**
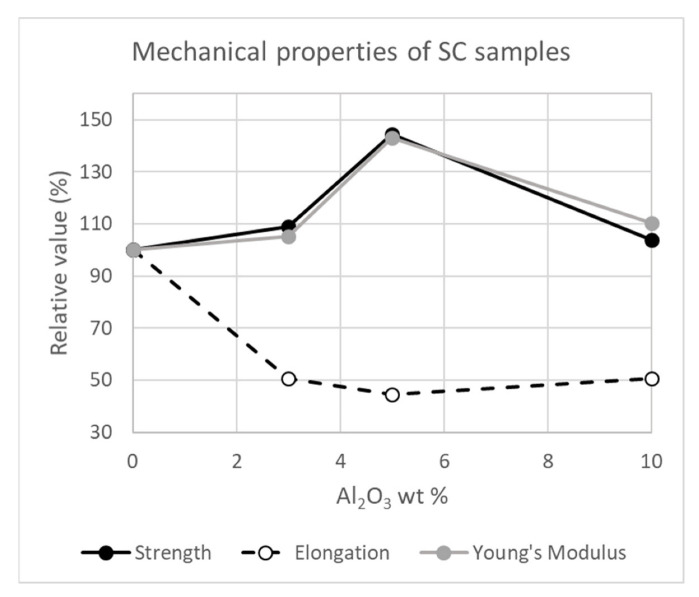
Relative elastic modulus, elongation at break, and tensile strength of the solvent cast samples as measured by tensile tests, taking as reference the sample that does not contain alumina.

**Figure 7 polymers-14-05033-f007:**
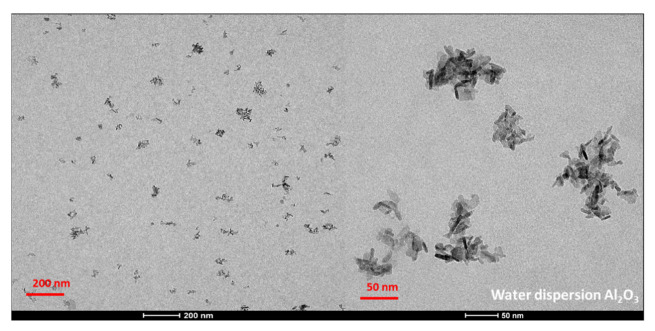
TEM micrographs of the reinforcing nanowire used (Al_2_O_3_).

**Figure 8 polymers-14-05033-f008:**
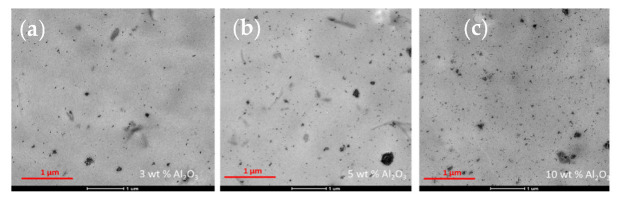
TEM micrographs for PHBH nanocomposites containing (**a**) 3 wt.%, (**b**) 5 wt.% and (**c**) 10 wt.% of alumina nanoparticles.

**Figure 9 polymers-14-05033-f009:**
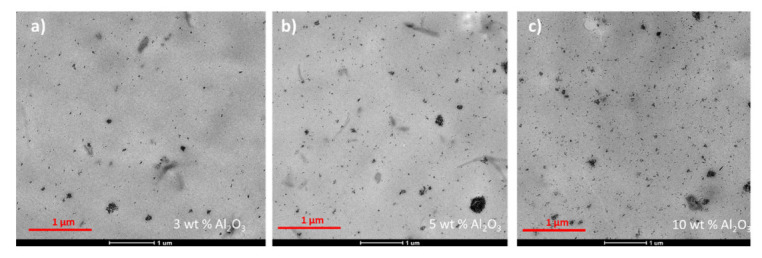
Binarized TEM micrographs for PHBH nanocomposites containing (**a**) 3 wt.%, (**b**) 5 wt.% and (**c**) 10 wt.% of alumina nanoparticles.

**Figure 10 polymers-14-05033-f010:**
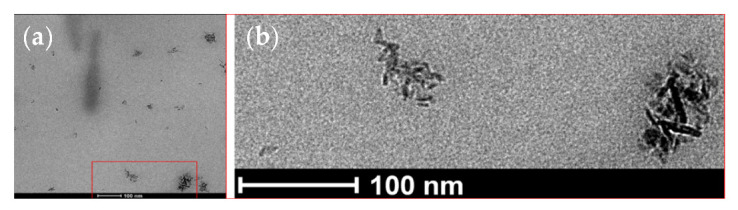
(**a**) TEM micrograph showing the morphology of the aggregates found in the 5 wt.% nanocomposite and (**b**) magnified image of the area marked in red.

**Figure 11 polymers-14-05033-f011:**
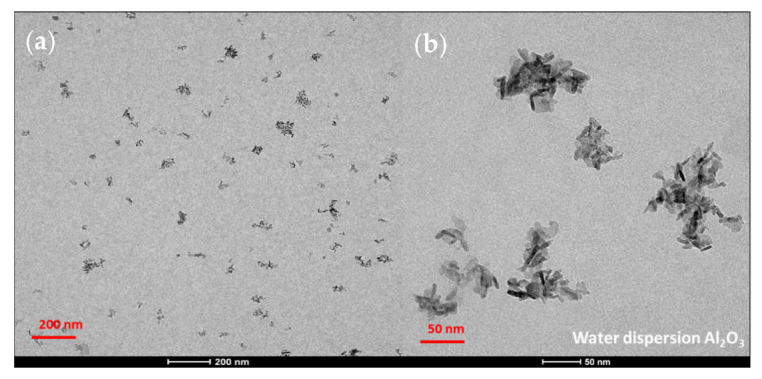
TEM micrographs for the 3 wt.% nanocomposites obtained (**a**) by solvent casting and (**b**) by melt blending.

**Figure 12 polymers-14-05033-f012:**
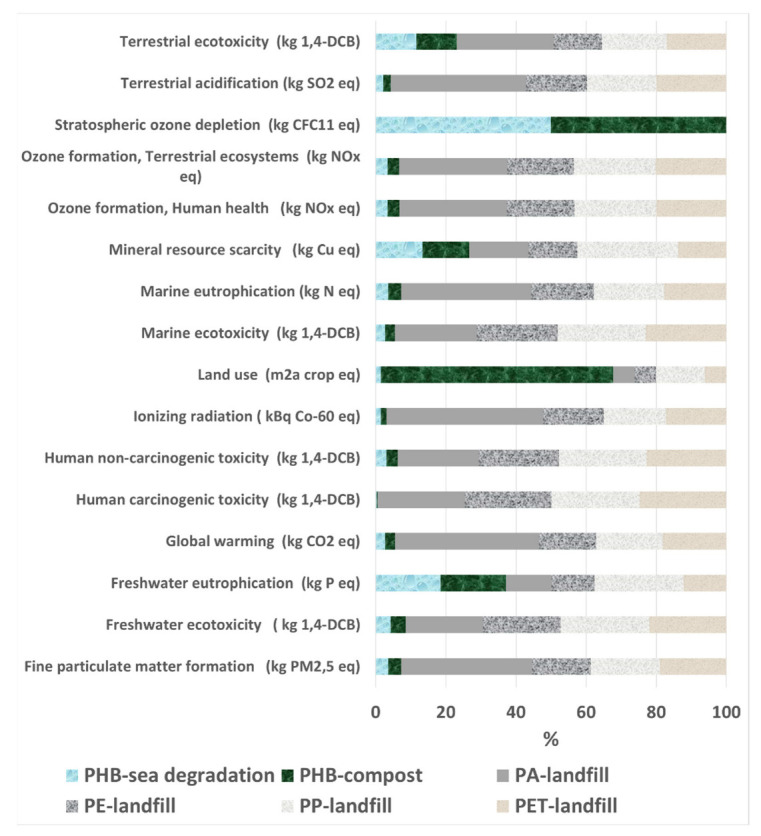
Relative contribution of cases “a”, “b” and “c” (%) to the quantified environmental impacts obtained using Recipe 2016 Midpoint.

**Figure 13 polymers-14-05033-f013:**
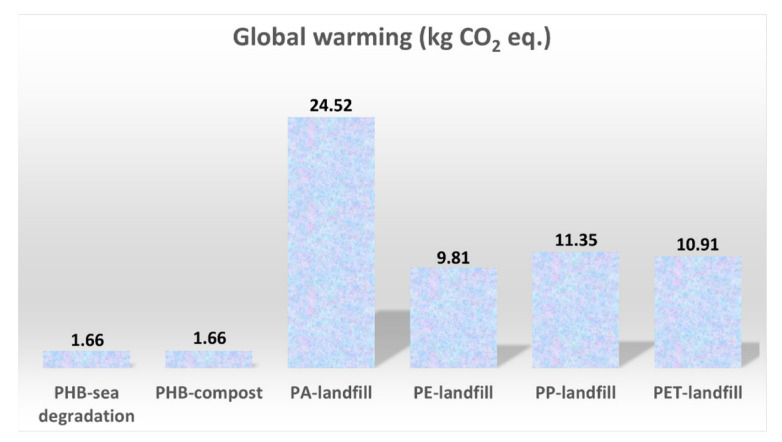
Global warming category for the studied cases “a”, “b” and “c”.

**Figure 14 polymers-14-05033-f014:**
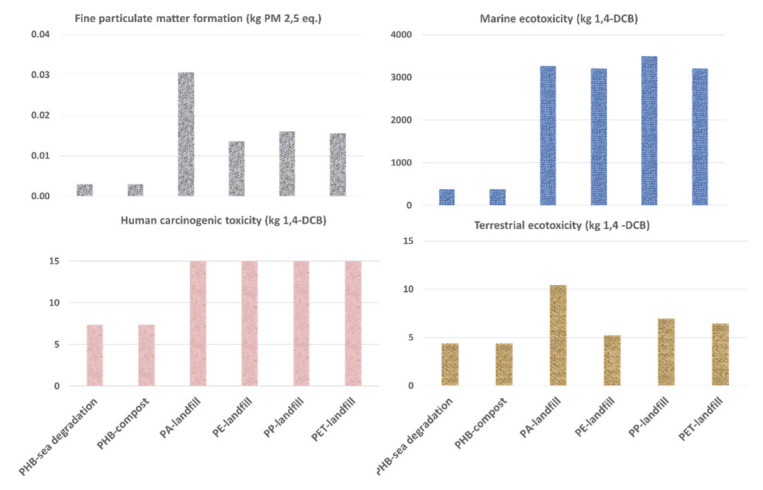
Fine particulate matter formation, marine ecotoxicity, human carcinogenic toxicity and terrestrial acidification categories for cases “a”, “b” and “c”.

**Figure 15 polymers-14-05033-f015:**
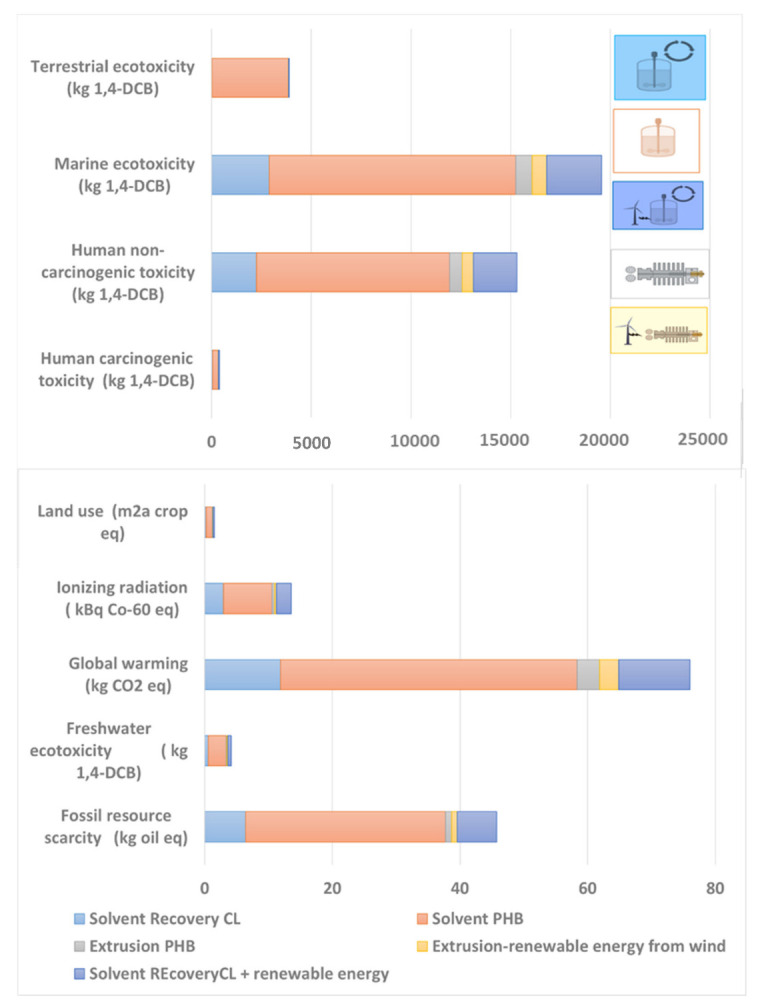
Contribution of each case “d” to “h” to the total of the selected environmental impact categories.

**Figure 16 polymers-14-05033-f016:**
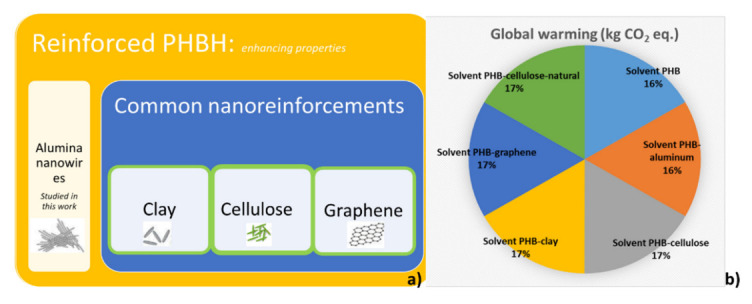
(**a**) Visual summary of the sensitivity analysis conditions to measure the effect of the different reinforcing particles (case “i”), and (**b**) the corresponding contribution of each case to the total global warming.

**Table 1 polymers-14-05033-t001:** Elastic modulus, elongation at break, and tensile strength of the solvent cast samples as measured by tensile tests.

Al_2_O_3_ (wt.%)	Strength (MPa)	Young’s Modulus (MPa)	Elongation at Break (%)
0	7.9 ± 0.9	580 ± 50	6.3 ± 0.7
3	8.6 ± 0.8	610 ± 40	3.2 ± 0.1
5	11.4 ± 0.9	830 ± 60	2.8 ± 0.2
10	8.2 ± 0.9	640 ± 80	3.2 ± 0.4

**Table 2 polymers-14-05033-t002:** Number of measured particles, particle density, average particle size (area) and the corresponding standard error of the PHBH nanocomposites prepared by melt blending as a function of the alumina content, as computed by Automated Image Analysis.

wt.% Al_2_O_3_	Number of Measured Particles	Particle Density (No. of Particles/µm^2^)	Average Size (nm^2^)	Standard Error of the Size (nm^2^)
3	3405	36	189	3
5	7934	61	173	2
10	17,103	133	168	1

**Table 3 polymers-14-05033-t003:** Green House Gas (GHG) emissions, Acidification Potential (AP) and Eutrophication Potential (EP) results of previous studies on PHB production (all data are reported per kg of PHBH). The data obtained in the current study have been added in the last row.

	Reference	GHG (Kg CO_2_-eq.)	AP (Kg SO_2_-eq.)	EP (Kg P-eq.)
1	Gerngross (1999) [29]	-	-	-
2	Akiyama et al. (2003) [23]	−0.24 to 0.82	-	-
3	Akiyama et al. (2003) [23]	0.48 to 1.39	-	-
4	Kim and Dale (2005) [30]	1.72 to 192	2.14 moles H^+^ eq.	1.9 g N-eq.
5	Kim and Dale (2005) [30]	−1.15 to −1.19	0.81 moles H+ eq.	1.1114 g N-eq.
6	Kim and Dale (2008) [31]	−2.3	-	-
7	Harding et al. (2007) [32]	1.96	24.9 g SO_2_ eq.	5.19 g PO_4_^−3^-eq.
8	Yu and Chen (2008) [33]	0.49	-	-
9	Rostkowski et al. (2012) [34]	9.42	92.5 moles H+ eq.	1.06 g N-eq.
10	Kendall (2012) [35]	3.4 to 5	16 to 28 g SO_2_ eq.	0.54 to 5 g PO_4_^−3^-eq.
11	I.K. Kookos et al. (2019) [36]	4.406	21.72 Kg SO_2_ eq.	1.18 Kg P-eq.
12	This work (production + processing + EoL)	1.66	0.006 Kg SO_2_ eq.	0.006 Kg P-eq.

## Data Availability

The data presented in this study are available on request from the corresponding author.

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
