# Peer review of "Sustainable PHBH–Alumina Nanowire Nanocomposites: Properties and Life Cycle Assessment"

_polymers, 2022, doi:10.3390/polym14225033_

Round 1

Reviewer 1 Report (Previous Reviewer 1)

Well Done!

Line 418, it is figure 12.

Author Response

Thank you very much for the review.

Reviewer 2 Report (Previous Reviewer 2)

This word presents the preparation of poly(3-hydroxybutyrate-co-3-hydroxyhexanoate)-alumina nanowire composites by solvent casting and melts mixing. Mechanical properties, microstructural analysis, environmental impact, and thermal stability have been studied. Alumina is applied for the first time in combination with PHBH. The results allow the comparison of the qualities of the composites with others, prepared with conventional polymers.

The word is well written and the discussions are based on relevant results.

Author Response

Thank you very much for the review.

Reviewer 3 Report (New Reviewer)

 In this work the authors studied the use of alumina nanowires to generate PHBH-alumina nanocomposites, modifying the properties of PHBH to improve its usability. Solvent casting and melt blending were used to produce the nanocomposites. Then, their physicochemical properties and aquatic toxicity were measured. Finally, LCA was used to evaluate and compare the environmental impacts of several scenarios relevant to the processing and end-of-life (EoL) conditions of PHBHs. I think the study is complete and is interesting. However, there are some revisions as follow:

1)      The organization of the manuscript should be concise, for example, some parts can be moved to supporting information.

2)      There are some grammatical problem, for example,”...by solvent cast-99 ing and melt mixing…”should changed to  “… by solvent casting and melt mixing method” in page 3.

3)      Some of the words in the figures are small for reading, please the authors update them.

4)      There are many format mistakes in the references part.

5)      Regarding to films materials, some new porous thin films are recommended to be the citations, e.g. Layer-by-Layer Grafting Dye on Enantiopure MOF Thin Films for Circularly Polarized Luminescence Amplification”, Chinese J. struct. Chem., 2022, 41, 2209074-2209079; Surface-Oriented Assembly of Cyclodextrin Metal-Organic Framework Film for Enhanced Peptide-enantiomers Sensing, CCS Chem. 2022, 4, 3472–3481; Tunable Chiroptical Application by Encapsulating Achiral Lanthanide Complexes into Chiral MOF Thin Films, Nano Res. 2021, 15, 1102–1108.

Author Response

Thank you for the review.

Please find attached the responses.

Dr. Maider Iturrondobeitia

This manuscript is a resubmission of an earlier submission. The following is a list of the peer review reports and author responses from that submission.

Round 1

Reviewer 1 Report

Please find the recommendation in the file.

Reviewer 2 Report

This work presents a study to prepare Poly(3-hydroxybutyrate-co-3-hydroxyhexanoate) alumina as nanowire nanocomposites. The authors study the use of alumina nanowires to generate PHBH alumina nanocomposites. This strategy allows the application of the Poly(3-hydroxybutyrate-co-3-hydroxyhexanoate) in replacement for petrochemical polymers. In contrast, raw Poly(3-hydroxybutyrate-co-3-hydroxyhexanoate) presents limited stiffness, a tendency to crystallize, and low resistance to thermal degradation, as nanowire containing alumina these difficulties can be overcome. The PHBH – alumina nanowire composites were prepared by solvent casting and melt mixing.

Although the work presents important aspects in terms of the proposed system, characterization techniques used, and even in the interpretation of the results, the way the text is formatted leaves something to be desired.

As an example, in Materials and Methods, some information should be presented in another way. For instance, the molar mass of Kaneka PHBH is given as 0.6X106 (line 107), and it should be given as 6X105 g/mol. On line 117 the information about the percentual composition of the samples is not referring any input (is difficult to understand).

Description of the ISO14040 should not appear in this form (between lines 150 and 168). In my opinion, it should only be cited, without explanation in the text. They remain not clear, the goal, the scope, and the inventory declared in the text. This section may be rewritten. The inventories declared in the text (Tables 1 to 4) are in fact the inputs used to prepare the samples. Perhaps this presentation does not need to be in the form of tables.

The Results must be presented as a function of the technique described in the Materials and Methods, and not just about certain properties, such as “Evolution of crystallinity over time” (this was determined by DCS, a technique described in the former section).

Several other problems related to the structure of the article could be mentioned. Finally, I suggest a complete revision of the main text for resubmission.

Reviewer 3 Report

In this manuscript, PHBH/Al-nanowires composites were prepared using melt blending and solvent casting method. Thermal properties and mechanical properties have been investigated. Morphological study using TEM presents some existing aggregates of Al-nanowires. Environmental impact Assessment by Life Cycle Assessment have been studied.

However, the manuscript is more like a “technical report” than a scientific paper. For instance, the authors should present the original TGA curves or at least offer some deviation bars for Fig. 3 and Fig.4. And the word “temperature” for the Y-axis in Figure 4 really lacks the basic understanding of a TGA curve. Which temperature does it mean?

Fig.5, no error bars? Only one sample have been tested? Especially when the aggregations have been observed   

Table.5, the average size of the particles are exactly 120,100,200? Without any deviation range? The value are just too neat to believe.

The section 3.5 was obtained using Recipe 2016 Midpoint E methodology, where no description on the methodology and analysis at all… Purely descriptive comparison between numbers is really beyond my understanding on scientific papers.